



# Wine must yields as indicators of May to July climate in Europe, 1416–1988

Christian Pfister[1], Stefan Brönnimann[1], Laurent Litzenburger[2], Peter Thejll[3], Andres Altwegg[4], Rudolf Brázdil[5], Andrea Kiss[6], Erich Landsteiner[7], Fredrik Charpientier Ljungqvist[8], Thomas Pliemon[9]

[1]Oeschger Centre for Climate Change Research, University of Bern, Bern, Switzerland, Institute of Geography, University of Bern, Bern, Switzerland.
[2]Associate Researcher, Université de Lorraine, CRULH, F-54000 Nancy, France.
[3]Danish Meteorological Institute DK-2100 Copenhagen O, Denmark.
[4]Independent Researcher, CH 5200 Brugg, Switzerland.
[5]Institute of Geography, Masaryk University, and Global Change Research Institute of the Czech Academy of Sciences Brno, Brno, Czech Republic.
[6]Research fellow at the Institute of Hydraulic Engineering and Water Resources Management. Technical University of Vienna, Austria.
[7]Univ.-Professor for Social and Economic History, University of Vienna, Austria.
[8]Professor of History, in particular Historical Geography, University of Stockholm, Sweden.
[9]Postdoctoral fellowship at the Fredy and Nadine Herrmann Institute of Earth Sciences, Hebrew University, Jerusalem, Israel.

*Correspondence to*: Christian Pfister (christian.pfister@unibe.ch). Stefan Brönnimann stefan.broennimann@giub.unibe.ch

**Abstract.** Narrative historical records of wine production in Central Europe date back to 1200. The extent to which such data can be used as proxy data for summer temperatures is being explored. Here, we investigated taxes paid to the authorities in the French-Luxembourg Moselle region, Germany and the Swiss Plateau over the last few centuries drawing on 11 regional series from the early 15th century to 1988. We detrended the data to compensate for longer-term biases. The homogenised series were merged into three supra-regional series: (1) the Moselle series, starting in 1416 and consisting of data from the former city-republic of Metz (France) and the Grand Duchy of Luxembourg; (2) the series for Germany, starting in 1511 and mainly originating from the former city-state of Heilbronn; and (3) the third series, starting in 1529 and representing production on the Swiss Plateau. The residuals of the supra-regional yield series were averaged, divided into seven classes, and multiplied by five quality classes. Yield quality indices (YQI) varying between 35 (large and excellent) and 1 (small and undrinkable) significantly correlated with temperatures between May and July. Regression analysis of the composite series revealed that yield and quality primarily depend on the climate conditions from May to July as well as on those in June of the previous year. Crops with a YQI > 28 (rated "good" by traditional winegrowers) were related to above-average May–July temperatures, early grape harvest dates and high tree-ring maximum latewood density resulting from frequent anticyclonic weather situations. Crops with YQI > 10 could not be uncritically assigned to cold summers since winter, and spring frosts often reduced yields without affecting quality. Severe crop failures sometimes triggered witch hunts. In summary, narrative evidence on wine production allows reconstructions based on tree-rings to be specified and verified.

**Keywords**

Vine must quantity, vine must quality, May to July temperature, frost impacts, witch hunts





## 1. Introduction

In chronicles and other sources dealing with the living conditions in pre-industrial Central Europa, wine is almost always on par with the staple food grain (Pfister and Wanner, 2021). It was simultaneously a religious symbol, an everyday drink and a luxury commodity. During the celebration of the Lord's Supper, red wine was used to symbolize the blood of Christ. Light Table wine served as an everyday drink, since clean water was not available in sufficient quantities prior to the 19th century (Toma, 2017). Quality wine aged for several years was a luxury commodity sold at a high price and traded over long distances (Landsteiner 2004). Up until the time of the viticultural crisis in the late 19th century, production was predominantly yield-orientated. According to vintage ratings, a "good harvest" was synonymous with a rich one.

There is now evidence of viticulture in Central Europe since Roman times (Matheus, 2025). It was mostly practised monoculturally in small vine gardens. In fact, viticulture required tilling the soil with hand tools and tending individual plants by skilled workers (both men and women). Though viticulture occupied only a small acreage, it produced a higher monetary value per unit area compared to arable products such as grain, albeit at the cost of unpaid family labour or high wage bills (Landsteiner, 2004). As late as 1930, about 3300 hours were required annually to cultivate a mere one hectare of vines in Switzerland (Altwegg and Pfister, 2025; Brazdil et al., 2019). The vine varieties remained stable until the phylloxera crisis, which started in the late 19th century and led to the replanting of vineyards with grafted vines on American rootstocks resistant to phylloxera. On this occasion, new, more productive varieties were often introduced, resulting in higher yields and better quality (Ollat et al., 2025).

Historical climatology aims to combine the narrative approach of the historical sciences with the quantitative approach of the natural sciences (Pfister and Wanner 2021). Wine must constitutes an unusually rich climatic source. Records relating to vine provide three climatic proxies, namely information on the start date of the grape harvest, the harvest quantity and the sweetness of the crushed grapes. Quality is least affected by human intervention before the harvest date, when the yield becomes most dependent on human activity. Gregory V. Jones and environmental scientist Robert E. Davis demonstrated that much of the annual variation in harvest date, yield and sugar content is controlled by a few large-scale weather conditions: Cyclonic weather patterns with strong winds and cold fronts reduce the quantity and quality of the must and delay the ripening of the grapes, while warm, high-pressure conditions allow early, extensive, high-sugar crops to ripen (Jones and Davis, 2000). Brázdil et al. (2008) suggested that narrative vintage ratings on wine must quality and quantity can provide a high-resolution summer temperature proxy for the period preceding the availability of grape harvest dates (GHD). For example, when grapes did not ripen in 1258 (a year without summer) after the explosion of the Indonesian volcano Samalas (1257), grapes were collected in sacks and baskets (Hertzog, 1906). In the hot summer of 1293, in contrast, "excellent wine grew in abundance" (Bassermann Jordan, 1907). So far, harvest date and vine must quality have been studied in the literature from a historical perspective (Labbé et al., 2019; Pfister et al., 2024).



This study aims to examine the extent to which narrative and accounting data on both crop size and sugar content may provide a proxy for spring and summer temperatures before the period for which GHDs are available. The paper ends with the implementation of quality-related restrictions in production after 1988.

## 2. State of research

German physician and natural scientist Gustav Schübler (1787–1834) can be considered the father of oenological climatology (Loose, 2022). In 1831, he compiled GHDs, fiscal series of wine must yields, wine prices and data on must quality for previous centuries to investigate "whether the climate of Germany has changed or remained the same for centuries, about which more detailed observations with physical instruments are still lacking […]". To this end, he came to relevant results (Schübler, 1831). However, his research fell into oblivion. Subsequently, grape harvest data and wine quantity and quality were analysed separately until recently.

The focus of nascent historical climate research was initially on GHD. Swiss physicist Louis Dufour (1832–1892) standardized four GHD series according to the Gregorian calendar and combined them into 30-year periods. In connection with results from the Ice Age research available at the time, he presented a comprehensive study of climate history (Dufour, 1870). Thus, he probably initiated a nationwide survey of GHD in French archives (Angot, 1885). On this basis, French historian Le Roy Ladurie (1971) reconstructed summer temperatures during the Little Ice Age (LIA) in connection with the fluctuation of the Alpine glaciers. Labbé et al. (2019) presented the first source-critically verified long time series from 1354 to 2018 for Beaune (France).

Wine must yields and wine quality were studied according to climatic aspects. After the forgotten pioneering work of Gustav Schübler, further wine chronologies were compiled; however, this was done without the critical evaluation of sources. In 1905, Historian August Hertzog (2006) published a comprehensive catalogue for the Alsace and Moselle region. After a decade of warm summers (1943–1952), botanist and oenologist Karl Müller (1881–1955) followed with a catalogue for southern Germany compiled from chronicles (Müller, 1953). Rima (1963) went one step further in his statistical search for periodicities in the time series of wine must yields and must quality at Johannisberg castle (Table 1-7).

Pfister (1981) and Trachsel (2009) examined the long yield series from the Swiss Plateau covering the period between 1529 and 1966, according to the best practice of historical climatology, using multiple regression based on instrumental measurements. They found that almost two-thirds of the variation in wine must yields can be explained by temperature conditions in early and mid-summer (June to July), where the June and July temperatures of the previous year account for almost a quarter of the total.

Lauer and Frankenberg (1986) investigated the relationship between wine must yield and quality in the Johannisberg castle vineyard series (Table 1-7) using multiple regression in conjunction with principal components analysis. They used this model



to interpret source-critical narrative data from the 17th to 19th centuries. In summary, their study concludes that large wine
must yields are indicative of warm and sunny weather with less drought in the early to mid-summer phase of the harvest year.
Molitor et al. (2016) interpreted a Luxembourg wine chronicle from 809 to 1904 that contained uncritical narrative information
on the quality and quantity of annual crops (Anonymous, 1937). The authors calculated the Heliothermal Index, i.e. the sum
of daily temperature maxima from April to September for the period between 1854 and 1885, which they related to narrative
reports on the quantity and quality of the grape harvests in the above-mentioned calibration period. From this, they calculated
the April–September temperatures for the period since the middle of the 15th century. This revealed significant relationships
between the narrative data on quality and the Heliothermal Index, while the information on quantity did not improve the result.
Lorusso (2013, 2018) used 26 time series from France, Germany and Switzerland to show that top-quality wine musts are
associated with early harvests, while acidic wines are associated with late harvests. Landsteiner (2004) dealt with the socio-
economic dimension of past wine must production in Austria.

Ljungkvist et al. (2025) examined the long vine yield series from Remich and Grevenmacher in Luxembourg (Yante, 1985) to
investigate the impact of volcanic eruptions on vine must production. They found a highly significant volcanic signature that
was stronger than the signal found in tree-ring-based reconstructions from Central Europe. The authors encouraged further
compilation and analysis of additional wine production series, given that such series contain unique historical, biological and
climatic information not found in other proxy types.

**3. Sources**

Mainly white grape varieties particularly resistant to adverse weather conditions were cultivated in our study area according
to Volk's (2009) comprehensive bibliography. Accounts of ecclesiastical and secular institutions contain annual revenues of
wine must paid in the form of tithes and rents. Much of them were spent in the form of wages in kind and pensions. The owners
of the vineyards, the clergy, the bourgeoisie and institutions such as monasteries and hospitals were situated in the cities, where
most of the wine was consumed.



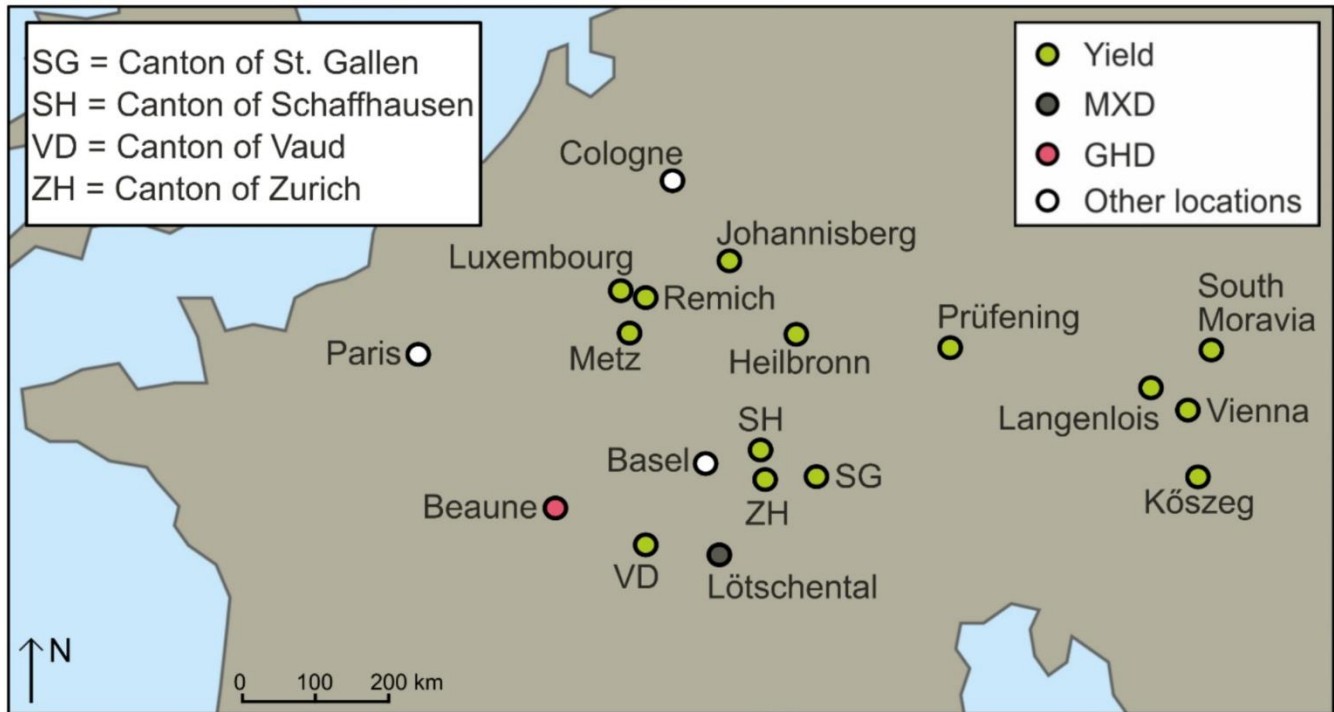

**Figure 01: Spatial layout of locations.**

The locations Bordeaux, Grevenmacher (situated close to Remich), Swiss Plateau, Lake Biel, Canton Aargau, Canton Thurgau
and Canton Valais are not shown for clarity.

Here, we shed new light on long series set up in the wine-growing city-state towns of Metz and Heilbronn. Moreover, an early
short series referred to the Bavarian monastery of Prüfening (today within the city of Regensburg). Yields from wine tithes are
documented in the accounts of the Swiss city-republics of Bern, Zurich, Schaffhausen and St. Gallen (vineyards in the upper
Rhine valley) (Pfister (1984). Krack (2019) and Niedermayr (2020) extracted time series of area yields from the Vienna city
hospital accounts between 1535 and 1682. A series of area yields from Langelois in Lower Austria was provided by Erich
Landsteiner. In connection with the French Revolution (1789), many institutions were dissolved, and their data dried up. It
was not until the late 19th century that the emerging nation states began to record wine production as part of official agricultural
statistics (Mitchell, 1992; Luxembourg, 1990; Ritzmann-Blickenstorfer, 1996).
Leased vineyards also act as a reliable source of information. Usually, the yields were shared between landowners and their
tenants. Since most of them had neither vine presses nor cellars to process and store their share of the harvest, they sold it at
institutionally prescribed prices (Pfister et al., 2024) so that they could buy grain and other products on the market for
themselves (Volk, 2009). The most reliable information is provided by long-term accounts of private estates. The 250-year-
long timeseries of the Johannisberg castle vineyard in the German state of Hesse is unique in this respect (Table 1–7)






**Table 1. List of sources.** Alt means Altitude, Lat means Latitude, M means Missing, Pri means Private, R means Residuals in percentage, Sh means Share of production, and St means national statistics.

| Src # | Location | Alt m | Lat | Type | Period | N | M% | References |
|---|---|---|---|---|---|---|---|---|
| 1 | Metz  (Fr) | 209 | >49° | Tax | 1416-1788 | 228 | 31 | Litzenburger 2015 |
| 2 | Remich (Lu) | 160 | >49° | Sh | 1444-1786 | 243 | 27 | Yante 1985 |
| 3 | Luxembourg (Lu) | 160 | >49° | St | 1839-1988 | 106 | 4 | Statec 1989 |
| 4 | Prüfening (De) | 337 | >49° | Sh | 1511-1530 | 20 | 0 | Weber 2023 |
| 5 | Heilbronn (De) | 160 | >49° | Sh | 1519-1921 | 403 | 27 | Dürr 1926 |
| 6 | Germany (pre 1991) | | | St | 1846-1988 | 135 | 5 | Mitchell 1992 |
| 7 | Johannisberg (De) | 90 | >49° | Pri | 1719-1950 | 7 | 0 | Rima 1963, Staab et al. 2001 |
| 8 | Swiss Plateau 1 | >400 | >47° | Sh | 1529-1797 | 269 | 0 | Pfister 1984, Table X |
| 9 | Swiss Plateau 2 | >400 | >47° | R | 1798-1966 | 169 | 0 | Trachsel (2009) |
| 10 | Swiss Plateau 3 | >400 | >47° | St | 1967-1988 | 23 | 0 | BLW 1967-1988 |
| 11 | Vienna (At) | 190 | >48° | Sh | 1535-1682 | 148 | 10 | Krack 2019, Niedermayr 2020 |
| 12 | Langenlois (At) | 219 | >48° | Sh | 1648-1748 | 101 | 0.5 | Stadtarchiv Langenlois, Bücher Serie 44/12 ff. |
| 13 | Köszeg (Hu) | 275 | >47° | Sh | 1723-1970 | 232 | 6 | Kiss (2011) |
| 14 | South Moravia (Cz) | 237 | >47° | Sh | 1692-1921 | 129 | 0 | request data from Brázdil |

From 1416 to 1988, 11 regional wine must yield series were compiled from the French-Luxembourg Moselle region, parts of Germany and the Swiss Plateau. The cities of Metz, Remich and Heilbronn are located above 49 N, while Vienna is situated at 48 N; the Swiss Plateau is located slightly farther south (Table 1).

The Metz regional series (Table 1–1) meticulously documents the extent of wine must production between 1416 and 1788 on the territory of the former city-republic, which is now part of France (Litzenburger, 2015). A tax of 23 deniers (approximately

1.4 grams of silver) per cuve (4.8 hectoliters) was levied at the conclusion of the grape harvest after St. Martin's Day (November 11), following an inspection of the new must stored in the cellars (Metz Archives). The vineyards were owned by members of the city's patriciate, ecclesiastical institutions and numerous small producers. Most of the wine was consumed locally in and around the city.

A comparison of the yields from 1473 to 1475 with those of the small town of Remich (Fig. 01, Table 1–2) reveals a significant

disparity, likely attributable to the siege laid on the city by Duc Nicolas I de Lorraine-Anjou in 1473. Consequently, the percentage values of Remich were employed for the corresponding years in the nearby Metz series (Fig. 01). Likewise, the





values for the period between 1538 and 1561 were absent from the archives after Metz was conquered by the French (Wikipedia, Histoire de Metz, https://fr.wikipedia.org/wiki/Histoire_de_Metz, accessed 12/27/2024). They were supplemented by the respective percentages of the Remich series, as well as further gaps in the years 1500 to 1502, 1521, 1530, and 1574.

The conquest of the city by French troops in 1646 during the Thirty Years' War (1618–1648) decimated the population (Trapp et al., 2021). From then on, there was a shortage of skilled labour. Starting in 1671, crops were only sporadically recorded. In 1678, the records stopped altogether. Fifteen years later, the records resumed, but yields were at a lower level, which persisted until the end of the Ancien Regime" in 1788.

The regional series from Remich (1444 to 1786) (Table 1–2) covers wine must production in the small town of the same name

in the district of Luxembourg. The corresponding measure of capacity – called Aimes – contains 158 litres. The Counts of Luxembourg received the ninth part of the harvested wine must from their estates there. After the harvest, inspectors went from cellar to cellar, as in Metz, to register the newly stored must. The vine was delivered in the castle to the counts (Yante, 1985).

Since the independence of the Grand Duchy of Luxembourg in 1839, the annual yields of this country situated near Metz (Fig.

01) were systematically recorded, with the quality measured in Oechsle degrees (Table 1–3). For the missing war year, 1944, the value of the Johannisberg castle series (Table 1–7) was used. Until 1918, the Grand Duchy was part of a customs union with the German Empire, which meant that wine was in high demand. Vineyards were tripled in size without regard for quality. From 1907, phylloxera spread almost unchecked. After the dissolution of the customs union in 1918, phylloxera-resistant and more productive vines were introduced in the 1920s and 1930s, which enabled a transition to quality production. (Massard,

175  2007).

The Prüfening regional series (Table 1–4) derives its nomenclature from the former Bavarian Benedictine monastery, now located in a suburb of the city of Regensburg. In the villages of Matting and Oberndorf, south of the city, the monastery was involved in the export-oriented wine industry. The produce from these vineyards was shared between the monastery and the tenants. Weber (2023) published the corresponding values for the years 1454 to 1457 and 1511 to 1530.

The Heilbronn regional series (Table 1–5) covers the years 1519 to 1803 and 1826 to 1921. After the expulsion of their sovereign, Duke Ulrich von Württemberg, the fathers of the current Free Imperial City of Heilbronn, levied a tax on the wine must yield beginning in 1519. Following the grape harvest, the number of "supply trips" was registered at the three city gates. A supply trip was equal to 3 hl of crushed, unpressed grapes (Schübler, 1831). The three-volume chronicle of Heilbronn lists the number of "supply trips" registered at all city gates. The document also contains information on quality and prices, as well

as references to weather extremes. The chronicler Friedrich Dürr based his information mainly on contemporary observations of schoolmasters (Heuss, 1950; Dürr, 1926). The above sources were destroyed in the bombing of Heilbronn on 4 December 1944. These documents were preserved for future generations largely due to the efforts of Dürr. However, with the incorporation of Heilbronn into the Electorate and, later, the Kingdom of Württemberg in 1803, the records were complemented by the Johannisberg series (Table 1–7). From 1827 to 1921, Dürr quotes production data from Heilbronn based

on the statistics of area yields in Württemberg. In 1881, the vineyards covered an area of 476 hectares (Pfaff, 1865).. The



Heilbronn wine statistics, covering a total of 380 years, represent the longest oenological time series known to date. They include information on quantity and quality as well as references to meteorological and exogenous factors.

The regional series for "Germany" is indicated in Table 1–6). From 1846 to 1988, vineyard area and wine must yields are displayed in (Mitchell, 1992), from which average area yields (hl/ha) were calculated. The absence of yield values for the war

and post-war years (1941 to 1947) was compensated for by the area yield series of the Johannisberg castle estate (Rima, 1963), which is the longest known so far.

Given the considerably longer-term differences in average yields, linear trends were calculated on a section-by-section basis for some regional series under the assumption that the annual residuals mainly represent short-term, primarily meteorological influences (Table 2).


**Table 2. The sections used to calculate regional trends**

| Series | | Section 1 | Section 2 | Section 3 | Section 4 |
|---|---|---|---|---|---|
| Metz | | 1461–1537 | 1561–1671 | 1693–1788 | |
| Remich | 1444–1570 | 1571–1603 | 1604–1647 | 1648–1680 | 1742–1786 |
| Luxembg | 1839–1957 | 1958–1988 | | | |
| Heilbronn | 1519–1568 | 1569–1688 | 1689–1803 | 1804–1871 | 1872–1921 |
| Germany | 1879–1958 | 1959–1988 | | | |

The regional series Swiss Plateau 1 (Table 1–8) is based on an estimation of wine must production between 1529 and 1797, using the vine tithe, the second most important feudal tax after the grain tithe (Pfister, 1984, Annex). The grape varieties largely

coincided with the linguistic and cultural boundaries. The white Chasselas grape was widespread in French-speaking western Switzerland, while traditional white varieties such as Elbling and Räuschling were common in the German-speaking cantons further east. Following the Reformation, the ecclesiastical estates in the Protestant cantons, along with their vineyards, became the property of the secular authorities. During the grape harvest, either the tithe payers poured the harvested grapes into an official wine press or a sworn overseer ensured that the 10th bucket was always emptied into an official tithe barrel (Pfister,

1981). After the dissolution of the Ancien Regime in 1798, tithes were abolished by canton in the following decades (Dubler, 2015). The Swiss Central Plateau regional series is divided into 10 sub-periods according to their different regional composition (see Table 2). A separate linear trend with residuals has been calculated for each canton.

**Table 3. Composition of the regional series Swiss Plateau by canton. Legend:** AG, later Canton Aargau; NE-Bi, Region Lakes Biel and Neuchâtel; SG, Upper Rhine valley (Altstätten, Canton St. Gallen); SH, Total production Canton Schaffhausen; TG, later Canton Thurgau; VD, later Canton Vaud; ZH, Canton Zurich (Pfister, 1984: Tab. XX Anhang).





| Subperiod | Tithe series | |
|---|---|---|
| 1529–1530 | SG, ZH | 2510 |
| 1531–1532 | SG | 1619 |
| 1533–1546 | SG, **ZH, NE-Bi** | 2702 |
| 1547–1566 | SG, ZH, NE-Bi, **VD** | 3505 |
| 1567–1631 | SG, ZH, NE-Bi, VD, **AG** | 3347 |
| 1632–1637 | ZH, NE-Bi, VD, AG | 3490 |
| 1638–1644 | **SG**, ZH, NE-Bi, VD, AG, TG | 3716 |
| 1645–1737 | ZH, NE-Bi, VD, AG, **SH**, TG | 7456 |
| 1738–1769 | ZH, NE-Bi, AG, SH, TG | 6759 |
| 1770–1797 | ZH, NE-Bi, **VD**, AG, SH, TG | 8076 |

A jump in the composition occurred in 1645 with the introduction of the Schaffhausen wine tax. From 1738 to 1770, data on the later Canton Vaud are missing from the compilation. The tithes documented in the six cantons of the Swiss Plateau (excluding Schaffhausen) come from a catchment area of approximately 5000 km², corresponding to a volume of 80760 hl between 1770 and 1797. Repeated war-like events caused significant damage in Metz and Heilbronn, while Switzerland – apart from the rapid conquest by French armies in 1798 – was only involved in four short civil wars (1531, 1656, 1712 and

1847), which meant that grapes could be harvested and stored without interruption year by year.

In contrast to the tithe series, a total of 16 series of area yields were compiled for the Swiss Plateau. They were derived from the financial records of leased vineyards owned by private individuals and official institutions (Pfister, 1984; Table 3). In most cases, the yield was shared between the leaseholder and the owner. Some records specify in detail the costs of cultivation, the frequency of fertilizing and the periodic rejuvenation of the vines. From 1533 to 1839, the name of the tenant, the surface of

the plot and the yield were annually recorded in the accounts of the city of Zurich. These records occasionally include comments on the condition of the vines, when those were affected by adverse weather. The total amount of tithes agrees largely (72%) with those of yields per area between 1533 and 1797 (Pfister, 1981). The regional series Swiss Plateau 2 from 1798-1966 (Table 1–9) builds on the regional series Tithe Swiss Plateau 1 (Table 1–10), employing data from the same cantons as well as from Swiss Plateau 3 (Table 1–10), extending from 1967 to 1988.

The supplementary evidence allows for assessment of the spatial range of the yield data. The series for Vienna (Table 1–11) is based on the accounts of the city hospital, which was a municipal institution responsible for the accommodation and care of the poor, the elderly, orphans, the sick and pilgrims. It managed a wine estate with an area of 90 to 100 hectares using paid labour (Niedermayr, 2020). However, the estate was seriously affected during the Ottomans' siege of Vienna in 1683 (Krack, 2019). The supplementary evidence from the lower Austrian town of Langenlois between 1648 and 1748 is based on the

accounts of the city hospital (Table 1–12). The supplementary evidence for the town of Kőszeg situated in Western Hungary




(Table 1–13) presents further information from the *Book of Vinesprouts* starting in 1740. This unique document details the measured annual size of vine sprouts on April 24<sup>th</sup> as a phenological proxy for the onset of spring, together with information on quantity and quality (Kiss et al., 2011) (Table 1–13). The data on quantity (Table 1–13) are listed in terms of an index from 5 (outstanding quantity) to 1 (complete crop failure). The supplementary evidence for the Czech historical region of Southern

Moravia (Fig. 01) presents annual yields in five classes for the period between 1692 and 1921 (Table 1–14), compiled from different places (Brázdil et al., 2008).

## 4. The formation of the supra-regional yield series

The 11 regional series were combined into three supra-regional series: Moselle, Germany and Swiss Plateau.

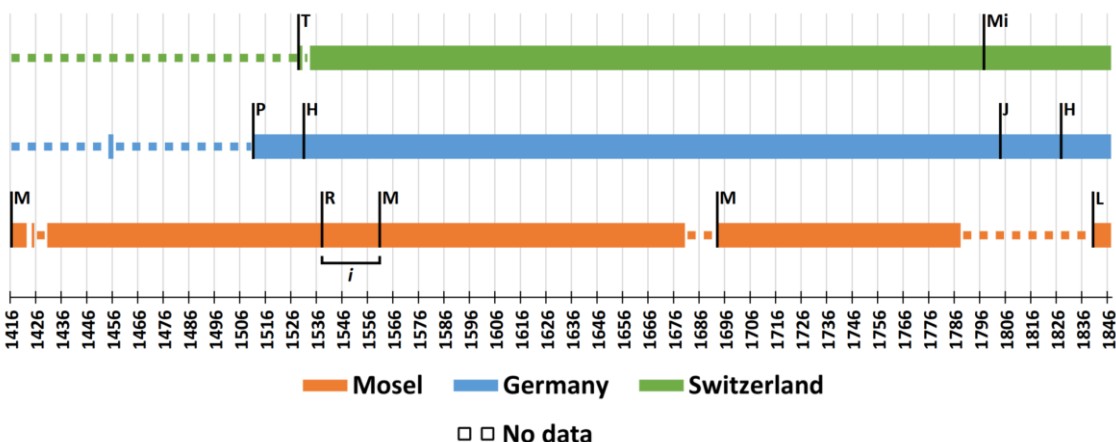

**Figure 02: The composition of the three supra-regional series. Legend: T...Tithe yields; Mi…Swiss Plateau; P…Prüfening; H…Heilbronn; J…Johannisberg; M…Metz; Mi…Metz interpolated; R…Remich; L…Luxembourg**

The supra-regional series Moselle comprises the two regional series Metz (Table 1–1), supplemented by Remich and Luxembourg (Table 1–3).

The supra-regional series Germany comprises the regional series Prüfening (Table 1–4), Heilbronn (Table 1–5), Germany (Table 1–6) and Johannisberg (Table 1–7).

The supra-regional series Swiss Plateau comprises the three regional series Swiss Plateau 1 to 3 (Table 1-8-10).




**Table 4. The correlation between yield series. Legend:** At Austria; Cz Czech Republic; CH Switzerland; De Germany; Lu Luxembourg; Hu Hungary. **Italics:** Supplementary evidence. **Data:** Southern Moravia: Brázdil et al., 2008.

|  | Moselle | Germany | CH Plateau | Germany | Luxembg |
|---|---|---|---|---|---|
|  | 1416–1788 | 1519–1921 | 1529–1988 | 1846–1988 | 1839–1988 |
| Heilbronn De 1519–1921 | 0.47 | — | 0.56 | **0.67** | **0.4** |
| Remich Lu 1444–1786 | 0.59 | 0.47 | 0.43 | — | — |
| CH Plateau 1528–1988 | 0.5 | 0.56 | — | **0.58** | **0.4** |
| Germany 1922–1988 | — | 0.67 | 0.58 | — | **0.62** |
| *Joannisberg De 1719–1950* | *0.58* | *0.51* | *0.53* | *0.68* | *0.42* |
| *Vienna At 1535–1683* | *0.29* | *0.22* | *0.27* | — | — |
| *Langenlois At 1648–1748* | *0.39* | *0.37* | *0.46* | — | — |
| *Moravia Cz 1692–1921* | *0.6* | *0.38* | *0.42* | — | — |
| *Köszeg Hu 1727–1970* | *0.4* | *0.1* | *0.3* | — | — |

As shown in Table 4, the Pearson correlation coefficients decreased with growing distance between the series. In fact, the coefficients between the series Moselle, Germany and the Swiss Plateau are still significant with the supplementary series Vienna, Langenlois, Köszeg and Southern Moravia at a distance of about 1000 km. The correlations between wine yields thus had a large-scale representativity. In contrast, the values between the Heilbronn and Swiss Plateau yield series were surprisingly low. This may be partly due to the dangerous fungus Downy Mildew introduced from the United States (Fontaine et al., 2021).

Fig. 03 to 05 present the three super regional series. Visual indications of severe frost in winter and spring are displayed in the lower part of the figures, while large harvests (> 30%) above the mean are displayed in the upper part.







**Figure 03: The supra-regional Moselle series (1416–1988).**

The first segment of Fig. 03 shows the wine tax yield within the geographical boundaries of the City Republic of Metz (see Table 1–1). After a period of predominantly good harvests from 1416 to 1425, yields declined until 1446 due to recurrent severe winter and spring frosts (Camenisch et al., 2016). After two substantial harvests in 1447 and 1448, yields remained below average until 1470, largely due to repeated cold summers caused by volcanic eruptions in the 1450s (Sigl et al., 2015). During the warm years (1472 to 1474), three large yields were recorded; meanwhile, yields between 1481 and 1495 fluctuated

widely due to erratic weather patterns. A decline until around 1530 was followed by a long period of stagnation that turned into a decade of frequent crop failures in 1586. Yields fluctuated above average between 1603 and 1636. Then they fell sharply, possibly because of the plague and the Thirty Years' War (1618–1648). In 1646, Metz was occupied by French troops, which might have caused a further slump in yields. From 1672 to 1693, records on yields are missing altogether. Disregarding a large





crop yield in 1706, yields were mainly below average until 1717. A sequence of warm summers shaped the period from 1718

to 1728, with the yield in 1718 achieving 400% of the average. A shorter run of large harvests was registered from 1779 to

1781. Spring frosts were frequent in the 1430s and the periods from 1561 to 1576, 1586 to 1602 and 1651 to 1671. Remarkably,

few frost events were recorded from 1531 to 1560, 1606 to 1625 and 1718 to 1739. The second part of the supra-regional

Moselle series showed deviations from the mean yields in the Grand Duchy of Luxembourg from 1839 to 1988 (see Table 1–

3). Excluding a record harvest in 1875, yields fluctuated around the average until 1893 and then fell to a low point in 1913,

partly due to infections with the downy mildew fungus (Fontaine et al., 2021). Spring frosts were frequent between 1913 and

1928. The switch to phylloxera-safe vines in the 1930s, supported by the state, led to an increase in yields and quality (Massard

2007). In the post-war period, the extreme conditions experienced in 1949 and 1950, as well as the poor harvest of 1956, which

was caused by winter frost and a cold summer, are noteworthy. A bumper harvest was registered in 1982.


**Figure 04: The supra-regional Germany series (1511 to 1988).**



Figure 04 refers to the rent paid to the Prüfening monastery (Table 1–4) until 1518. From 1519 to 1803 and from 1826 to 1921,
it applies to the Heilbronn regional series (Table 1–5). The period from 1804 to 1825 is covered by the Johannisberg castle
series (Table 1–7). The subsequent dataset presents the calculated average area yield data for the former Federal Republic of
Germany from 1922 to 1988 (Table 1–6).

Disregarding the initial dip in harvest around 1520, harvests were mostly favourable up until 1552. This upswing was driven
by moderate grain prices, relatively high wine prices and a growing workforce. About 10,000 hectares of new vineyards were
planted in the Duchy of Württemberg between 1514 and 1566 (Alber, 2016). Sharecropping offered the land-poor lower classes
the opportunity to get a living from a small plot of land, subject to favourable weather conditions. After 1552, yields fell to a
low in the cold early 1570s, only to rise to a short peak in 1584. A succession of late frosts, harsh winters and cold summers
for the next two decades resulted in poor harvests. Grain prices rose relative to wine prices in the context of recurrent crop
failures. Consequently, many tenants were unable to meet their families' needs through the sale of wine (Landsteiner, 1999).
A substantial harvest in 1605 initiated a third upward trend, which culminated in the double bumper harvests of 1630 and
1631. Yields fluctuated around average until 1687. Cold seasons were frequent over the next two decades. However, yields
somewhat recovered until 1708, when they turned low again until 1716. The 12 years between 1718 and 1729 stand out for
their unique sequence of bumper harvests. The mostly low yields of 1740 and 1747 were followed by small to average harvests
until 1779 and by large harvests in 1781, 1783 and 1784. The town was repeatedly occupied by French troops, first in 1688
during the Nine Years War (1688–1697), then from 1734 to 1736 during the War of the Polish Succession (1733–1738) and
finally in 1794 and 1799 during the First and Second Coalition Wars, respectively (1792–1797; 1798–1802) (Dürr, 1926). The
Johannisberg castle series from 1803 to 1825 includes a series of poor crops during the cold phase from 1812 to 1817. The
years from 1826 to 1868 are characterized by high yield variability. Despite nine bumper harvests, only a small amount of
wine could be stored in seven years. Following this period, the range of yield fluctuations narrowed, although the losses in the
years between 1886 and 1910 might have been due to the downy mildew disease (*Plasmopara viticola)* (Fontaine et al., 2021).
In the post-war period, there were poor harvests in 1956 and during the wet summer of 1980. Two bumper harvests in 1982
and 1983 led to overproduction. Spring frosts were frequent from 1586 to 1611, 1773 to 1784 and 1828 to 1878. Further
records are missing for the remaining period.







**Figure 05: The supra–regional series Swiss Plateau (1529–1988).**

Figure 05 refers to the supra-regional series Swiss Plateau covering the period from 1529 to 1988. After a poor crop yield in 1529, yields rose to a bumper harvest in 1540 and then alternated above average until 1557. A longer-term decline in yields began in 1558, accelerating in the early 1570s. After a brief warm spell from 1580 to 1584, winter and spring frosts, superposed by cold summers, affected the vines until 1602. From 1603 to 1617, yields increased, culminating in a bumper harvest in 1616. Over the subsequent 13 years, yields mostly remained below average. Bumper harvests in the 1630s ushered in a period of great variability that lasted until 1688. The following decade was marked by severe frosts and cold summers with poor harvests. The period from 1718 to 1731 stands out with four bumper harvests and three above-average yield years. Between 1740 and 1811, above- and below-average harvests alternated, especially during a cold spell from 1769 to 1773. The early 19th century was characterised by a series of poor harvests between 1812 and 1817 and again between 1830 and 1833. Apart from a dip between 1888 and 1891, yields moved around average until 1908. Only three harvests were satisfactory between 1909 and 1928 due to the weather and downy mildew (*Plasmopara viticola*) disease (Altwegg, 2023). Apart from a severe crop failure



due to winter frost in 1956 and a severe spring frost the following year, yields fluctuated above average until 1988, including
two large harvests in 1982 and 1983. No further records of spring frost are available.

The supra-regional series share some common features, including a tendency for above-average harvests in the first half of the 16th century. In fact, summer temperatures in Central Europe from 1531 to 1540 were 0.25°C (± 0.49°C) above the mean for the period between 1961 and 1990 (Dobrovolny et al., 2010). Second, an accumulation of low and poor harvests from 1585 to 1602 is evident in all three series, when estimated annual temperatures in Central Europe were 1.2 °C (± 0.37 °C) below the
mean for the period between 1961 and 1990 (Dobrovolny et al., 2010). Third, the period from 1689 to 1705 was characterised by a comparable accumulation of poor and failed harvests. Annual temperatures in Paris were 0.7°C below the mean for the period between 1901 and 2000. Fourth, record harvests were observed between 1718 and 1729, when summers in Paris were 0.3°C above the average for the period between 1901 and 2000 (Rousseau, 2015). Fifth, yields remained consistently low during the frosty summers of 1813 to 1817. Sixth, in all three regions, yields generally declined between the 1890s and 1913,
partly due to climate and partly due to the downy mildew (*Peronospora viticola*) disease (Altwegg, 2023). Finally, large yields were registered in 1982 and 1983. Concluding on the magnitude of yield losses, severe frosts tended to cause more damage in the Mosel and Heilbronn regions located further north than in the Swiss Plateau.

In Metz and Heilbronn, warfare repeatedly caused considerable damage, while Switzerland, apart from its rapid conquest by French armies in 1798, was only involved in four brief civil wars. Moreover, total yields came from a much larger catchment
area than those in Metz or Heilbronn, which might have somewhat balanced out fluctuations . Finally, average yields ranging from 30 to 50 hl/ha on the Swiss Plateau (Pfister, 1981) were considerably higher than those in Johannisberg castle (23 hl/ha) (Table 1–7), Vienna and Langenlois (17.5 ha/hl) (Table 1–11 and 1–12). This difference may reflect a better supply of fertiliser to the Swiss vineyards due to nearby cattle grazing areas.

## 5. Statistical methods

To produce homogenised time series of wine must production, expressed as deviation in percentage from the long-term mean, we applied the following methods. In each interval $i$, we calculated the mean value of the must-series and called it $m_i$. In each interval, we fit a low-order polynomial (such as a straight line, or $2^{nd}$, $3^{rd}$, $16^{th}$, etc.) to the data using ordinary least squares. We used orthogonal polynomia since it increases numerical stability; there was a small noticeable effect on the results, e.g. fourth- and fifth-order polynomia, when we used our data. We subtracted the fit from the data, resulting in residuals $r_{ij}$, where
$i$ is the interval and $j$ is the year.

We calculated the ratio of the residuals to the interval's mean value and expressed it as a percentage:

$$S_{ij} = \frac{R_{ij}}{m_i} \times 100 \qquad\qquad (1)$$



We obtained wine must production data in an already contiguous series for the Swiss Plateau. For these series, a linear fit was used to remove the trend. We also obtained a set of series for southern Germany, where the former was detrended linearly and the latter with a fifth-order polynomial. A single series of wine must data from the Swiss Plateau covering just 1967–1990 was also processed using fifth-order detrending. Data from Germany for 1847–1988 showed a gap during the Second World War, but we made up for that using data from Johannisberg (covering 1933–1947) for the overlapping years 1933–1939. A linear transformation for the overlapping years was calculated, and the Johannisberg data for years 1940–1947 were transformed with it. Then, the filled-in data were tested for statistical similarity in the periods from 1920–1957 and 1958–1988 using the Kolmogorov–Smirnoff test. In detail, the null hypothesis that both datasets are drawn from the same population was tested. The result showed that the null hypothesis could be strongly rejected, implying that the data from the two periods were not drawn from the same population – that is, they were very different. We next detrended and converted the interval-mean to percentages using break years at 1879 and 1988. It seemed the resulting 46 series was not variance stationary, with a suggested break near 1950, and a similar procedure was followed for data from Switzerland (1837–1989), except there were no gaps to fill. Also, here, break years were specified as 1879 and 1958.

## 6. The formation of the yield quality index

The calibration method commonly used in palaeoclimatology is not appropriate for the climatic interpretation of wine must yields because large harvests are not always of good quality, and high-quality harvests are not always large. To some extent, this bias is due to legacy effects (Liu et al., 2025) connected to the formation of flowering buds in the summer before the harvest year. However, severe winter and spring frosts could also significantly reduce grape yields without affecting the quality.

The yield quality index (YQI) allows narrative vintage ratings, such as "much and good" and "little and sour", to be used as a proxy. For this study, the mean residuals of the three supra-regional yield series were averaged. The result was divided into seven classes according to size. The yield classes were then multiplied by the five quality classes established by Pfister et al. (2024) whereby the order was reversed from 5 (very good) to 1 (poor). YQI could thus take values from 35 to 1. Due to the larger number of yield classes, the crop data were given a higher weight in the multiplication than the quality data corresponding to the practice of historical vintage ratings. The GHDs available for the entire period were used for validation. Figure 06 provides a visual overview.



**Figure 06: YQI and GHD from 1416 to 1988. Red columns indicate a YQI > 20 and blue columns a YQI < 10.**





Figure 06 presents a visual agreement between YQI and GHD. The red columns show "good harvests" with YQI > 70. The
blue columns show harvests with YQI < 10. "Good harvests" were frequent between 1416 and 1425, 1471 and 1473, 1539 and
1540, 1677 and 1681, as well as 1857 and 1858. This relationship is broken down in more detail in Table 5.

**Table 5. Frequency of grape harvests according to YQI for 1416–1988. Legend**: YQI: Yield quality index; f: frequency; f
%: Relative frequency; GHD: Average of standardized GHD for Beaune (France) from 1354 (Labbé et al., 2019); T°C:
Deviation of May to July temperatures from the 1961–1990 average in Paris (1659 to 1988) (Rousseau 2014); MXD: Measured
Tree-ring Maximum Density (MXD) in Lötschental (Schweiz): Büntgen et al. (2006) (Fig. 01).

| Yield Class | | Quality Class | | f | f% | Ø GHD | Δ T°C | Ø MXD |
|---|---|---|---|---|---|---|---|---|
| 7 | Bumper crop | 5 | Excellent | 22 | 4 | -1.6 | 1.3 | 1.1 |
| 6 | Large | 5 | Excellent | 21 | 4 | -0.3 | 0.7 | 0.3 |
| 7 | Bumper crop | 4 | Good | 31 | 5 | -0.4 | 0.6 | 1 |
| 6 | Large | 4 | Good | 17 | 3 | -0.1 | 0.7 | 0.2 |
| 5 | Surpassing | 4 | Good | 12 | 2 | 0.1 | -0.4 | -0.3 |
| | | | | 103 | 18 | | | |
| 4 | Average | 4 | Good | 55 | 10 | -0.1 | 0.5 | 0.1 |
| 3 | Mediocre | 3 | Mediocre | 153 | 27 | 0.5 | -0.5 | -0.4 |
| 1-2 | Small | 1-2 | Sour | 262 | 45 | 0.8 | -0.9 | -1.3 |
| | | | | 573 | 100 | | | |

Table 5 shows that 18% of the grape harvests can be qualified as "excellent" or "good" and 10% as quantitatively above
average and of good quality; 27% were mediocre, and 45% "poor" and "sour" or "undrinkable". Qualitatively and
quantitatively "good" harvests went hand-in-hand with above-average summer temperatures, above-average to early GHD and
above-average maximum tree ring densities (MXD). However, this connection needs to be broken down in more detail (Table
6).







**Table 6. Range of YQI related to proxy data and May to July temperatures. Legend:** YQI: Yield quality index; GHD: Grape harvest dates for Beaune (Fr) (Labbé et al., 2019); MXD: Maximum tree ring density Lötschental (Büntgen et al., 2006); N: Number of cases; T°C: Deviation of May to July temperatures in Paris from 1961 to 1990.

| *YQI* | *GHD* | N | *MXD* | N | *T°C* | N |
|---|---|---|---|---|---|---|
| *>28:* | *-1.1* | *44* | *1.2* | *44* | *0.0* | *20* |
| *>25:* | *0.38* | *56* | *0.1* | *114* | *-0.4* | *56* |
| *< 10:* | *0.64* | *256* | *-0.6* | *252* | *-0.6* | *144* |


It can be concluded from Table 6 that years with YQI > 28 are, on average, related to early GHD, high MDX values and above the 20[th] century's average May to July temperatures. This result may be interpreted in terms of large-scale weather conditions according to the findings of Jones and Davis (2000) regarding the Bordeaux region in the 20th century. This relationship vanishes with YQI values below 28. Likewise, YQI values < 10 are on average connected to May to July temperatures below

the 1961–1990 average, late GHD and low MDX values.

## 7. The climatic model

Quantity, quality, and YQI can be very well modelled from climate data. We used temperature and precipitation data from the grid point 48.4 °N, 6° E from a monthly, global, climate reconstruction with monthly resolution (Valler et al., 2024) that is independent (i.e. it does not include the two-time series) and calibrated a model in the period 1781–1880 consistent with

previous work (Pfister et al., 2024).

A backwards variable selection approach was used, initially comprising all monthly temperature and precipitation values from April in the preceding year to August in the current year. However, only variables statistically significant at $p < 0.05$ for one of the three series were kept. The resulting model (Table 6) shows that quantity depends on the previous years' temperature (note the negative sign for April's temperature), while quality does not. Conversely, quality depends slightly negatively on

precipitation, while quantity does not. Overall, we find high explained variances, reaching 66% for quantity, 74% for quality and 75% for YQI. This confirms that all three series can serve as climate indicators.

**Table 7.** Coefficients (units per °C for temperature, units per $10^5$ mm/mon for precipitation), explained variances and adjusted explained variances of regression models for quantity, quality, and the YQI calibrated in the 1781 to 1880 period (bold: $p < 0.05$).





| 1781–1880 | Quantity | Quality | YQI |
|---|---|---|---|
| $T_{Apr-1}$ | -0.058 | -0.044 | **-0.523** |
| $T_{May-1}$ | **0.159** | -0.030 | 0.351 |
| $T_{Jun-1}$ | **0.444** | -0.025 | **1.139** |
| $T_{Apr}$ | 0.041 | **0.211** | **1.034** |
| $T_{May}$ | **0.034** | 0.261 | **2.038** |
| $T_{Jun}$ | **0.673** | 0.291 | **3.352** |
| $T_{Jul}$ | **0.395** | 0.158 | **1.699** |
| $T_{Aug}$ | 0.116 | **0.124** | **1.036** |
| $P_{Apr}$ | -0.121 | **-0.183** | **-1.175** |
| $P_{May}$ | **-0.186** | -0.049 | -0.728 |
| $P_{Aug}$ | 0.005 | **-0.202** | -0.536 |
| $R^2$ | 0.664 | 0.735 | 0.749 |
| adj. $R^2$ | 0.622 | 0.701 | 0.718 |

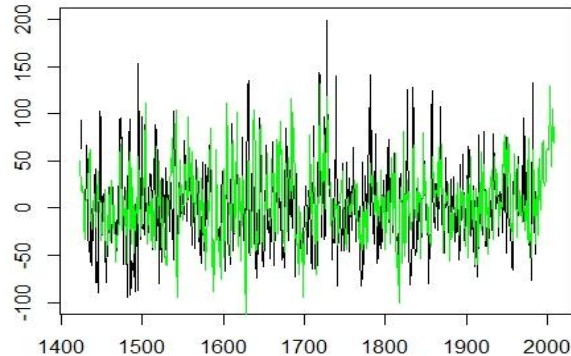


**Figure 07: Observed (black) and modelled (green) series of YQI.**

Figure 07 shows the modelled and observed YQI series from 1421 to 2008. For this figure, the modelled series was cut at the minimum and maximum values, i.e. at 1 and 35. The figure confirms the excellent agreement, with correlations of 0.87 in the

calibration period and 0.60 over the entire period back to 1421. The YQI could be constructed further back in time and then used for climate reconstruction.



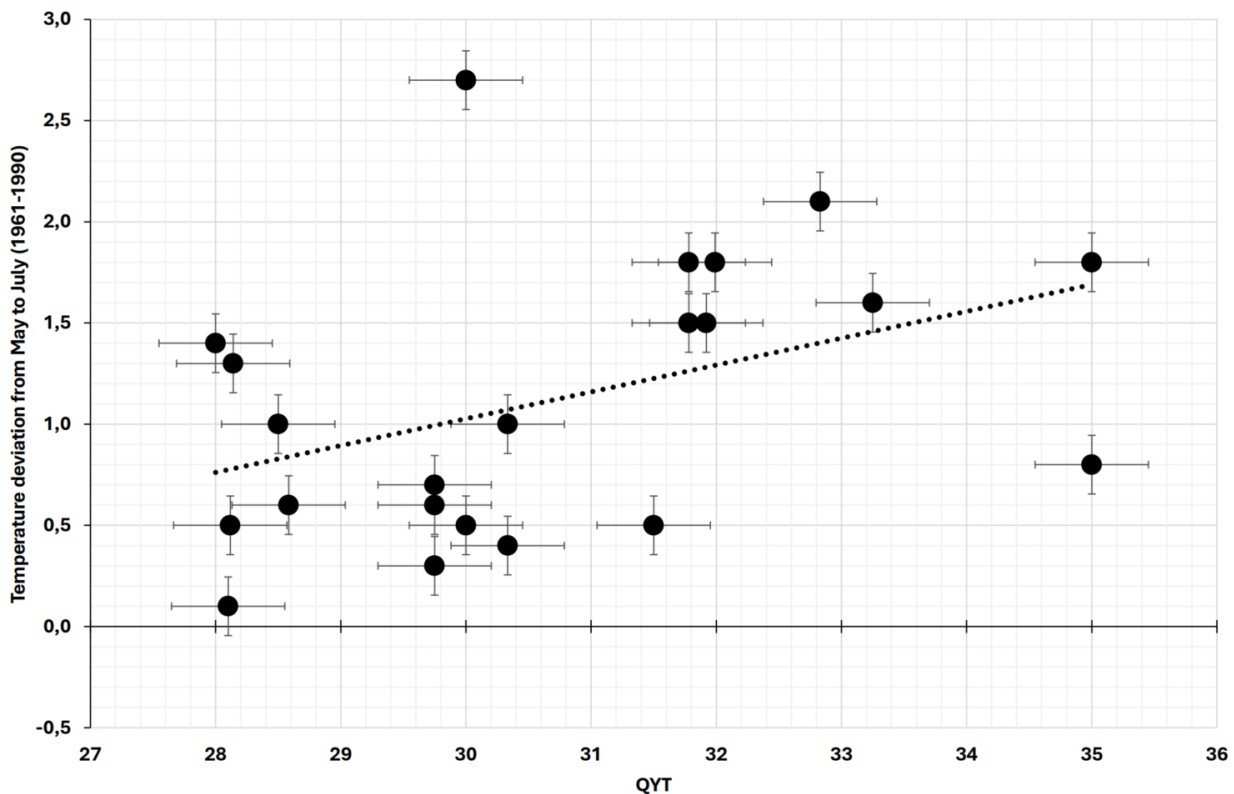

**Figure 08: Linear regression of YQI > 28 with May to July temperatures in Paris from 1658 to 1988.**

Figure 08 details the results of Table 6 regarding YQI > 28. It corresponds to a "good crop", according to the practice of

vintage ratings in the historical past. The linear regression of YQI > 28 with May to July temperatures shows a curve rising significantly according to y = 0,1326 x = 2,9516 and r = 0.41 (N =22).

Moreover, YQI values < 10 are significantly correlated with r = 0.28 (N = 144), but low values might also be due to severe winter or spring frosts in contrast to cold summers. Frosts in spring due to the advection of (sub)polar air can cause significant damage, particularly in years with advanced phenophases. For example, the highest economic loss of €3.3 billion caused by

late spring frosts (particularly to fruit and wine) occurred in Europe in 2017, with Italy, France, Germany, Poland, Spain and Switzerland being the most affected (Faust and Herbold, 2018). During the LIA, such calamities particularly affected marginal areas like Metz, Luxembourg and Heilbronn (Fig. 03 and 04). One of the worst calamities in the historical past occurred on 5 May 1517 (Gregorian calendar). In the morning, it was "hard frozen" in Metz with "deep snow". The leaves of the vines appeared "burnt". Based on reports from Alsace (Hertzog, 1906), this calamity affected regions from the upper Rhine Valley

(Fig. 09) to the Regensburg area (Table 1–4). It was a large-scale event. Moreover, when winter temperatures dropped below -15 to -20°C vine buds were destroyed during dormancy. A cold snap was particularly devastating after a mild period, drawing from the example of the well-known "millennium winter" of 1709. Warm and rainy weather with south-westerly winds



prevailed until 5 January, when temperatures during the following night plummeted far below freezing (Pfister and Wanner, 2021). Daily temperatures of -15°C were recorded over the next two weeks in Paris (Rousseau, 2024). The vines in Metz

recovered, even after two hard frosts in late spring, but they did not bear fruit (LePage, 1854). The situation in February 1956 was similar (Pfister and Wanner, 2021), resulting in the destruction of 45% of the vines in the Bordeaux region (Pérès, 2007). From this perspective, vintage ratings of "good" crops by contemporary chroniclers may have been interpreted in terms of proxy data for "warm" or "cold" summers. But before the availability of GHD, such reconstructions would need to be checked against tree-ring evidence.

**8. Socio-economic and cultural-historical aspects**

Vineyards were a worthwhile investment before the establishment of savings banks in the 19th century (Bartlome, 2014). On Lake Biel, they yielded a 5% interest during the 18th century (Pfister, 1981). Above-average grain and wine harvests provided good conditions for both tenants and the owners of vineyards. Fetching a good price, wine was in sufficient demand due to moderate grain prices. However, repeated bad grain harvests caused grain prices to rise faster than wine prices, which means

tenants bought less grain on the market with their share of the harvest (Landsteiner, 2004). In case of overproduction, there were insufficient barrels to store the abundant vine and consumers to cope with the surplus of wine despite low prices.

**8.1 Crop failures**

Wine prices tended to be negatively correlated with yields. However, a long-term increase in the price of wine encouraged a switch to substitutes. Beer was not suitable as a substitute for a long time. In the Middle Ages, it was made with herbal mixtures,

resulting in a cloudy, sweet, hard-to-store drink with a low alcohol content (Wikipedia, History of Beer, https://en.wikipedia.org/wiki/History_of_beer, accessed 17 April 2025). By the end of the 13th century, the addition of hops in northern German cities made beer tastier, longer lasting and exportable. Thus, hop beer became the standard (Landsteiner, 2004).

Harvest failures were hotbeds of cultural history. A hard frost in spring announcing impending disaster set vine-growing

communities under tremendous stress. Storm bells were rung in churches, and processions were held in supplication (Litzenburger, 2015). From the middle of the 15th century, such calamities became hotspots for witch hunts. Litzenburger (2015) cites an account by the chronicler Philippe de Vigneulles: "At 4 o'clock in the morning on May 1, 1456, the vines in Metz froze. People were totally distressed. A 16-year-old boy was arrested because he was seen in the company of witches and devils who allegedly belonged to a diabolical sect. They were said to have thrown the devil's tears into a well to cause

frost. Under pressure, the young man denounced a group of people who under torture named other 'co-conspirators'. Thirteen victims ended up at the stake. This example shows that the classical model of witch-hunting developed in the 15th century in viticultural areas. According to Litzenburger (2015), each crisis led to a conflict among members of the community. By getting





together against isolated victims, the community tried to regain its unity. It was up to the legal institutions to "cleanse" the social body of the threat.

The culture of remembrance (https://en.wikipedia.org/wiki/Culture_of_Remembrance) serves to maintain risk awareness. In the historical past, this objective was achieved through reports and illustrations in chronicles, or visual monuments such as flood markers (Pfister and Wanner, 2021). Likewise, a visual monument of risk awareness situated at the wall of the "House of the Bottle" in St. Gallen reminds the spring frost disaster in 1517 (Fig. 09).

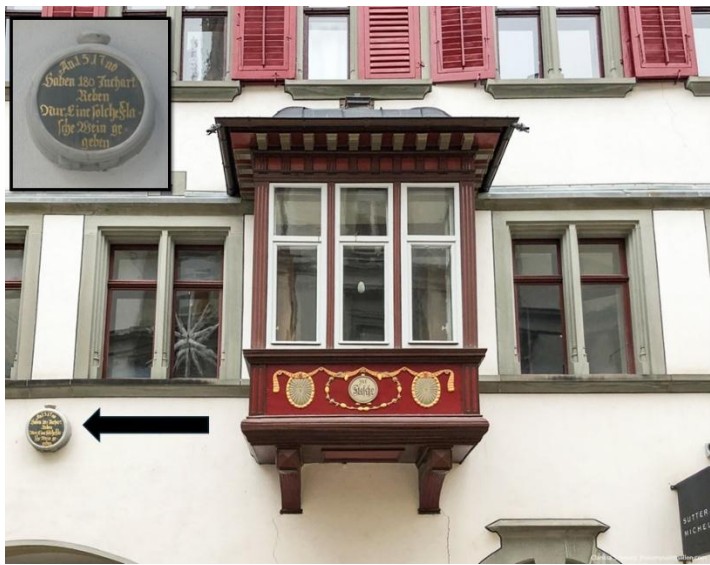

**Figure 09: Composite photograph of a memorial epigraph displayed at the 'House of the Bottle' Spisergasse 11 in St. Gallen (Switzerland). The inscription in German reminds of the failed grape harvest in 1517. Its translation reads: "In 1517, 180 Juchart of vines [7.2 ha], yielded only one bottle of wine" [in the St. Gallen Rhine Valley]. Photos Hans Fässler, St. Gallen.**

## 8.2 Overproduction crises

After two successive plentiful harvests, prices plummeted and barrels to store the bounty became short. In 1483 and 1484, many grapes were left hanging, and wine was given away "for God's sake" (Hertzog, 1906) (Fig. 10).



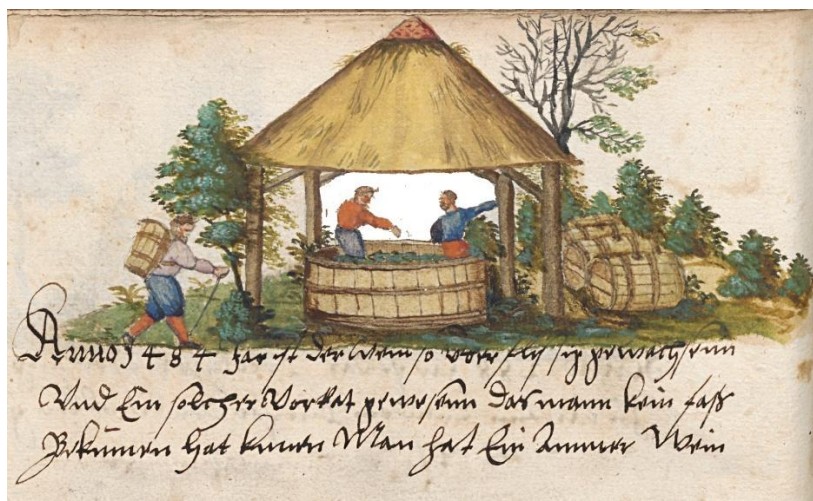

**Figure 10: The plentiful vine must harvest of 1484 in Nuremberg (Germany). This illustration within a chronicle reminds the bumper crop in 1484. The man coming from the left drags freshly picked grapes along. The man in the red coat crushes the ripe grapes with**
**his feet due to the lack of a press. The man in the blue coat points to the full barrels to the right, visualising the record harvest. Source: Stadtarchiv Stadt Nürnberg, Neubauersche Chronik (1601), code F 1 Nr. 42 S. 19.**

Likewise, the overabundant average quality wine obtained in 1539 was thrown into gutters in 1540 when a new, cheap, honey-sweet and highly alcoholic wine became available after the hot summer of that year. In Cologne, many people were
found "lying in the streets and behind hedges like pigs", according to the chronicler Hermann von Weinsberg (Pfister, 2017). Women in Heilbronn were forbidden from drinking wine, except during a period at the end of Lent. Husbands then had to drag their wive home in the evening (Weingärtner, 1962) During the warm summers between 1718 and 1727, the wine glut in Central Europe peaked, with seven large and above-average harvests. Chronicler Abraham Pagan disapproved of the fact that owners of vineyards in Lake Biel offered their wine for sale on the streets, even to children (Pfister, 1981). In France,
the planting of new vineyards was prohibited after the abundant wine harvests of the 1720's . (Lachiver 1988). Likewise, the double bumper harvests of 1982 and 1983 in Central Europe ended up in a glut. In  Siwss Canton Valais, wine was stored in private swimming pools and railroad tankers. Many large producers went bankrupt (Laessle, 2018). Disposing of the surplus cost the Swiss taxpayers millions of France and led the Federal Government to enact quality regulations and quantity restrictions (Altwegg, 2023). To relieve the European market, 120 million litres of wine were converted into industrial fuel
(Der Spiegel, 47/1983).

## 9. Conclusion

Narrative ratings of wine must quantity, along with quality information, provide a highly resolved summer temperature proxy for the period before the availability of GHD. This result was obtained from annual tax receipts on the Swiss Plateau as well as in the German city-states of Heilbronn and Metz (modern-day France) in connection with data from the Grand Duchy of



Luxembourg. The evidence was merged into a summary series. Due to the substantial effects of winter and spring frosts and the legacy effects of summer temperatures, the yield data could not be directly related to temperatures in the summer half-year. Evidence regarding the size of the grape harvests was instead merged with data on quality into a a yield-quality index YQI. This index reflects the practice of vintage ratings in the historical past, which prioritised quantity somewhat more than quality. The YQI is 70% correlated with the monthly temperatures between May and August, with a focus on May to July. In

contrast, low YQIs are correlated with low May to July temperatures, but this result may also include the effects of summer temperature legacy effects and winter and spring frosts. The YQI will have to pass the acid test in future studies in comparison to high resolution evidence from the archives of nature. It is expected that the available evidence will be used in follow-up studies on wine prices in the historical past.

**Competing interests:** The authors declare that they have no conflict of interest.

**Author contribution**: CP was responsible for conceptualisation, data curation, formal analysis, investigation, methodology, resources, validation and writing, and original draft preparation, as well as review and editing. SB was responsible for climatical analysis, modelling, validation, visualisation, writing, and review. LL took care of the graphics, supplied data and reviewed the draft. PT was responsible for the homogenization and statistical analysis.  AA provided a critical review of the

oenological arguments.  RB and AK provided supplementary data. EL provided supplementary data and a critical review. FL provided a critical review of the draft. TP provided a critical review of the draft.

**Acknowledgements**

- Dr. Hans Fässler St. Gallen

- Stadtarchiv Nürnberg (Germany), http://www.stadtarchiv.nuernberg.de

. Dr. Saskia.David-Gaubatz and Sabine Graham, Stadtarchiv Heilbronn (Germany) https://www.stadtarchiv.heilbronn.de/

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
