# Peer review of "Wine must yields as indicators of May to July climate in Europe, 1416–1988"

_EGUsphere, 2025_

## Referee Comment (RC2)

Title: Wine must yields as indicators of May to July climate in Europe, 1416–1988. Author(s): Christian Pfister, Stefan Brönnimann, Laurent Litzenburger, Peter Thejll, Andres Altwegg, Rudolf Brázdil, Andrea Kiss, Erich Landsteiner, Fredrik Charpentier Ljungqvist, and Thomas Pliemon.

MS No.: egusphere-2025-3242

MS type: Research article

Iteration: Initial submission

**1. GENERAL COMMENTS**

| Principal criteria                                                                                                                                                                                                  | Excellent | Good | Fair | Poor |
|---------------------------------------------------------------------------------------------------------------------------------------------------------------------------------------------------------------------|-----------|------|------|------|
| Scientific significance: Does the manuscript represent a substantial contribution to scientific progress within the scope of Climate of the Past (substantial new concepts, ideas, methods, or data)? | x         |      |      |      |
| Scientific quality: Are the scientific approach and applied methods valid? Are the results discussed in an appropriate and balanced way (consideration of related work, including appropriate references)?   | x         |      |      |      |
| Presentation quality: Are the scientific results and conclusions presented in a clear, concise, and well-structured way (number and quality of figures/tables, appropriate use of English langage)?          | x         |      |      |      |

**Presentation quality:** Regarding the quality of the writing in English, please ask confirmation from an English-speaking colleague (this is not my case).

**2. SPECIFIC COMMENTS**

- 1. Does the paper address relevant scientific questions within the scope of CP?
- 2. Does the paper present novel concepts, ideas, tools, or data?

YES

3. Are substantial conclusions reached?

**YES**

4. Are the scientific methods and assumptions valid and clearly outlined?

**YES**

5. Are the results sufficient to support the interpretations and conclusions?

**YES**

6. Is the description of experiments and calculations sufficiently complete and precise to allow their reproduction by fellow scientists (traceability of results)?

YES

7. Do the authors give proper credit to related work and clearly indicate their own new/original contribution?

YES

8. Does the title clearly reflect the contents of the paper?

VES

9. Does the abstract provide a concise and complete summary?

YES

10. Is the overall presentation well structured and clear?

YES

11. Is the language fluent and precise?

YES - If this question includes the quality of the writing in English, please ask an English-speaking colleague for confirmation (this is not my case).

12. Are mathematical formulae, symbols, abbreviations, and units correctly defined and used?

YES

13. Should any parts of the paper (text, formulae, figures, tables) be clarified, reduced, combined, or eliminated?

YES: enlarge figure 07, in order to improve its readability.

14. Are the number and quality of references appropriate?

**YES**

15. Is the amount and quality of supplementary material appropriate?

There is no supplementary material attached to this paper.

**3. TECHNICAL CORRECTIONS**

No corrections to be made.

**SUMMARY**

(version en Français) L'utilisation de données proxy liées à la viticulture n'est pas nouvelle, comme les auteurs le détaillent de façon précise avec les références bibliographiques appropriées. De ce point de vue, l'intérêt majeur de ce travail est d'avoir fait appel à l'utilisation de différentes sources de données liées à la vigne, et dont l'analyse combinée a produit des résultats aussi pertinents que passionnants.

Je souhaite aussi souligner la pertinence géographique des sites d'étude. En effet, l'intégration des régions de l'Est de la France à une étude sur l'Europe centrale n'est pas systématique d'une publication à l'autre, alors qu'elle se justifie par la régionalisation climatique de l'Europe. Ce choix est donc d'un intérêt tout particulier dans l'article de C. Pfister *et al*.

(*English version*) The use of proxy data related to viticulture is not new, as the authors detail in a precise way with the appropriate bibliographical references. From this point of view, the major interest of this work is to used different sources of data related to the vine, and whose combined analysis has produced results as relevant as exciting.

I also wish to emphasize the geographical relevance of the study sites. Indeed, the integration of the regions of Eastern France into a study on central Europe is not systematic from one publication to another, even though it is justified by the climatic regionalization of Europe. This choice is therefore of particular interest in the article by C. Pfister et al.

---

## Author Comment (AC1)

Revision Wine production and Climate 2025-2

Answers to Carlo Mateus

Title: I suggest slightly changing the title to reflect the contents of the paper clearly: Wine must yields as indicators of May to July climate in Central Europe, 1416–1988'

**Agreed**

Line 17: "This sentence requires a reference; therefore, it must be removed from the abstract and included elsewhere in the manuscript (e.g. Introduction). In addition, only data from 1416 is being used as a proxy.

**Solutions:**

- 1. The sentence was omitted, because narrative documentary records were not systematically included in the text. Rather, the value of documentary data for vine must production was investigated using quantitative records.
- 2. Line 17 was corrected as follows: The paper explores to which extent documentary records on wine production in Central Europe can be used as proxy data for summer temperatures.
- 3. Line 76: briefly specify the relevant results.

This study has shown that narrative documentary data on both crop size and sugar content may provide a proxy for spring and summer temperatures prior to the availability of instrumental measurements-.

**4. Final sentences**

In a follow-up study a reconstruction of summer temperatures based on narrative sources will be attempted for the period prior to the early fifteenth century. The YQI will have to pass the acid test in in comparison to high resolution evidence from the archives of nature. Moreover, it is expected that the available evidence will be used in follow-up studies on wine prices in the historical past.

It must be clearly stated why there is no analysis of data after 1988. What are the reasons? Data availability? It would be interesting to include the recent warmer decades in the analysis if data are available.

The following explication is added (lines 71-74:

The paper ends with the transition to the more rapid anthropogenic warming after 1988. As the world has warmed, the growing season has lengthened and phenological events have occurred earlier. Furthermore, mean quality has increased considerably (Altwegg 2023, Altwegg and Pfister, 2024) and yields would have risen steadily were it not for market-based control measures. Additionally, grape varieties better suited to the new climate are being cultivated (Neumann and Matzarakis, 2011; Holzkämper et al., 2013). Rising temperatures will not necessarily decrease the risk of late frosts. Water scarcity is likely to become a problem in some regions and an increase in the risk of pests is to be expected (Van Leeuwen et al., 2024). Therefore, data on wine production from after 1988 cannot be compared with data from previous centuries.

Make sure to add the references: Lines 39-40, 44, 57-58, 75-76, 536-537.

Line 39-40: Pfister and Wanner 2021 is quoted.

Line 44: Landsteiner 2004 is quoted in the references.

Line 44-45: "Up until the late 19th century, production was predominantly yield-orientated. (Altwegg 1979, Pfister 1981).

Line 57-58 ----the reference on line 59 should be Pfister et al. 2024

Line 75-76 The sentence refers to the quote from Schübler, 1831

In lines 495-496, 510 and 536-37 the references related to Wikipedia were replaced. The bibliography has been updated accordingly.

Section 3. Sources – It would be great to include a few figures of the original sources to highlight a few examples

Two examples are provided in the cited literature. The article is already close to the maximum length.

Section 5. Statistical methods. Please make sure to use a statistical textbook/peer-reviewed publications to add references to the used methodologies:Low order polynomial, ordinary least squares, orthogonal polynomials, Kolmogorov–Smirnoff test and detrending techniques. Adding references is important to ensure the description of experiments and calculations is sufficiently complete and precise to allow their reproduction by fellow scientists (traceability of results).

The section was revised.

The bibliography has been updated accordingly.

It would be great to cross-reference the results of the analysis of the historical documentary data with references on future effects of climate change on wine grape production in terms of quantity, quality, and resilience of the wine regions included in the study area.

An explication was provided in lines 71-74

**Technical corrections**

Line 38: Europe, not Europa. o.K.

Line 40: table, not Table. o-K

Line 265 (table): Luxembourg.----o.K-

The term Mosel designates the wine region, also in English, while the river is written Moselle.

Figure 8: Decimal points must be used instead of commas in the Y axis: The figure was corrected

I hope my comments are viewed as positively constructive and will assist the authors in enhancing their manuscript. I wish the authors well as they consider these comments and edit their manuscript for possible publication in the Climate of the Past.

The authors hope that you will understand our decision not to go into further detail about the events after 1988.